# Simulation of Folding Kinetics for Aligned RNAs

**DOI:** 10.3390/genes12030347

**Published:** 2021-02-26

**Authors:** Jiabin Huang, Björn Voß

**Affiliations:** 1Institute of Medical Microbiology, Virology and Hygiene, University Medical Center Hamburg-Eppendorf, Martinistraße 52, 20246 Hamburg, Germany; j.huang@uke.de; 2Computational Biology Group, Institute of Biochemical Engineering, University of Stuttgart, Allmandring 31, 70569 Stuttgart, Germany

**Keywords:** folding space, kinetics, position-specific abstraction, RNA, conservation

## Abstract

Studying the folding kinetics of an RNA can provide insight into its function and is thus a valuable method for RNA analyses. Computational approaches to the simulation of folding kinetics suffer from the exponentially large folding space that needs to be evaluated. Here, we present a new approach that combines structure abstraction with evolutionary conservation to restrict the analysis to common parts of folding spaces of related RNAs. The resulting algorithm can recapitulate the folding kinetics known for single RNAs and is able to analyse even long RNAs in reasonable time. Our program RNAliHiKinetics is the first algorithm for the simulation of consensus folding kinetics and addresses a long-standing problem in a new and unique way.

## 1. Introduction

The structure–function relationship is a generally accepted and often documented property of RNAs, be it non-coding, messenger, ribosomal, transfer or other RNAs. Structure formation is driven by thermodynamics with the goal to minimise free energy, but it is not guaranteed that thermodynamic equilibrium is reached. Biologically, this is not a problem because evolution selects by function and not by structure, but computationally this means that the biologically active structure is not the one we can predict by free energy minimisation. Insight can be gained by studying the folding kinetics, which allows for identifying comparably stable folding intermediates. Furthermore, in some cases, structural changes are essential for the function of an RNA. For example, a Riboswitch that binds the ligand it senses changes the structure in its expression platform, leading either to de- or increased expression of the downstream open reading frame (ORF). Similarly, the structure needs to fold back into the initial conformation upon ligand release to ensure correct function. These functionalities are encoded in the folding kinetics of the respective RNAs and, thus, efficient tools for kinetic studies are of large interest. The major problem in kinetic analyses is the extremely large state space, i.e., the enormous number of possible foldings, and the even larger number of possible transition paths between these states.

For short RNAs, complete enumeration and simulation at elementary step resolution is possible, although already computationally expensive. To study folding kinetics of longer RNAs, we need to cut down the computational demands. For this, one can either reduce the number of states and/or the number of transition paths under consideration. For the first idea, several approaches have been proposed: The *macrostate* approach introduced with treekin [1] considers solely local minima, which can be computed by barriers efficiently, at least for an energy range above the *mfe*, and their transition states (saddle points).

Another approach is to perform a stochastic simulation of folding trajectories. Starting from an initial state, e.g., the open structure, base pairs are added and removed in a stochastic fashion. Whether a transition to a candidate structure is performed or not depends on its free energy difference to the originating state. This approach is implemented in kinfold [2]. The main advantage is its speed and low memory footprint, but this is partly outweighed by the need to calculate many hundreds or thousands of trajectories. This is the only way to get an overview of kinetically preferred structures. This approach can also be used to sample folding trajectories between two predefined structures, which is helpful especially for long sequences.

Predicting folding trajectories between two predefined structures can also be done heuristically. Here, the idea is to allow only those moves that add or remove base pairs that are unique to either the start or end structure. One variant of this is the breadth-first-search (BFS) heuristic introduced in [3], which evaluates all neighboring structures that can be reached by adding or removing a non-shared base pair. From those, it keeps the *k* best and continues until the target structure is reached. Although, to the best of our knowledge, it has never been investigated thoroughly, this and related heuristics perform better the more similar the structures are. For dissimilar structures, this means that the prediction by these heuristics could be improved if reliable intermediate structures could be identified. Of course, it is not trivial to define reliability in this sense and further more to efficiently compute the reliable ones from the exponentially many intermediate structures.

One approach in this direction is shape abstraction [4], where structures are mapped to abstract representations, the so-called shapes. This mapping can be interleaved with the free energy optimisation, such that shapes, together with their representative structure, can be computed efficiently. The initial shape abstraction did not distinguish between equal shapes at different positions, e.g., a hairpin at the beginning or the end of an RNA. While this may have benefits for some applications it is not ideal for the estimation of folding trajectories. With RNAHeliCes, a position-aware structure abstraction was introduced and later also used to compute folding trajectories and to simulate folding kinetics. The corresponding algorithm HiKinetics [5] uses helix-index based abstraction to sample the structure space and the BFS heuristic to compute folding trajectories between the sampled structures. More specifically, the tool HiKinetics decomposed the folding space into disjoint classes, so-called *hishapes*, and calculated folding pathways between their representatives (the member with minimum free energy in each *hishape*) using HiPath2 to estimate energy barriers between *hishapes*. The energy barriers are further fed into Treekin to derive transition rates using Arrhenius’ equation.

Another problem in the simulation of folding kinetics is that the underlying thermodynamic model uses simplifications and approximations, such that the predicted structures might be reasonable under the model, but are unfavourable in nature. This is a general problem in RNA structure analysis and a widely adopted approach is to include evolutionary information, i.e., as in RNAalifold [6,7], RNAlishapes [8] and others. These algorithms start from multiple sequence alignments and combine free energy minimisation with the analysis of conservation and covariation of base pairs. This does not only improve prediction accuracy, but additionally reduces the size of the near-optimal state space compared to single sequence predictions. For the simulation of folding kinetics, the benefit is that fewer potential folding pathways have to be considered.

In this contribution, we describe RNAliHiKinetics, which merges abstraction based simulation of folding kinetics with alignment based structure analysis. We briefly introduce the conceptual approach and show applications in comparison to single sequence based methods.

## 2. Materials and Methods

### 2.1. Implementing RNAliHeliCes

Bellmans GAP, a 2nd generation language and system for Algebraic Dynamic Programming (ADP) [9], splits a DP algorithm into a grammar, describing the search space, and several algebras for scoring, candidate representation and abstraction. This concept makes it straightforward to design new algorithms by reusing grammars and algebras. Furthermore, Bellmans GAP supports single-track (e.g., a sequence) and multi-track (e.g., an alignment) input.

The grammar we use for RNAliHeliCes is the same as that for RNAlishapes [8], and was originally described in [10]. It is also known as *canonicals_nonamb* in the Haskell version of RNAshapes, or *MacroState* in [11]. In general, in Bellmans GAP, each piece of grammar can be applied to single-track or multi-track inputs without adaptations. However, syntactic filters, such as the basepairing filter described in the following, and algebra need to be adapted to the respective data structure. In the case of RNA folding, it is central to check that two bases (i,j) can form a valid base pair. For a single sequence (i,j)∈{(A,U),(U,A),(G,C),(C,G),(G,U),(U,G)} must hold, but for an alignment, positions *i* and *j* refer to columns, which may additionally contain gaps, where base pairs are not necessarily valid for all sequences. As already introduced in [8], we make use of a base pairing filter that checks the fraction of valid base pairs and drops candidates that do not meet a user-defined threshold, which is 0.5 by default.

Algebras for the pretty printing of structures in dot bracket notation and *hishape* mapping were taken from RNAHeliCes without modifications. In contrast, for the calculation of free energies, some adaptations were necessary. We implemented the same pseudo free energy scoring as in [7,8,12], which is a combination of mean free energy and a covariation contribution as shown in Equation (Equation 1).
(1)$$\Delta G_{i,j}^{\#}={\bar{\Delta G}}_{i,j}+\lambda \times C_{i,j}$$
where $\Delta G_{i,j}^{\#}$ is the pseudo free energy, ${\bar{\Delta G}}_{i,j}$ the mean free energy and $C_{i,j}$ the covariance score over all alignment columns at positions *i* and *j*. λ is a weighting factor to balance free energy and covariance contribution. The covariance score $C_{i,j}$ is calculated as
(2)$$C_{i,j}=V_{i,j}-\phi \times Q_{i,j}$$
where $V_{i,j}$ is a conservation score, $Q_{i,j}$ is a penalty, and ϕ a scaling factor. $V_{i,j}$ and $Q_{i,j}$ are computed as follows:(3)$$\begin{array}{cc} V_{i,j}= & \frac{1}{M}\sum_{1\leq k<l\leq M}\left\{\begin{array}{cc} h\left(s_{i}^{k},s_{j}^{k}\right)+h\left(s_{i}^{l},s_{j}^{l}\right) & \mathrm{if}\;\left(s_{i}^{k},s_{j}^{k}\right)\in \mathrm{BP}\;\mathrm{and}\;\left(s_{i}^{l},s_{j}^{l}\right)\in \mathrm{BP}\\ 0 & \mathrm{otherwise} \end{array}\right. \end{array}$$
(4)$$\begin{array}{cc} Q_{i,j}= & \frac{1}{M}\sum_{k=1}^{M}\left\{\begin{array}{cc} 0 & \mathrm{if}\;s_{i}^{k}=s_{j}^{k}=\mathrm{gap}\;\mathrm{or}\;\left(s_{i}^{k},s_{j}^{k}\right)\in \mathrm{BP}\\ 1 & \mathrm{otherwise} \end{array}\right. \end{array}$$
where BP = {(A,U),(U,A),(G,C),(C,G),(G,U),(U,G)}, *M* is the number of sequences in the alignment, and h(x,y) is the Hamming distance between bases *x* and *y*. $s_{i}^{k}$ is the *i*th base in row *k* of the alignment.

### 2.2. Implementing RNAliHiPath

Computing potential folding pathways between consensus structures of aligned sequences is the same as for single sequence structures because the used heuristics work on the structure representation and use the sequence only for energy evaluation. As a result, plugging in a function for evaluating the energy based on an alignment is sufficient to adapt the algorithm. We used the function energy_of_alistruct() from the Vienna RNA Package 2 for this purpose. The function performs the same computation as given by Equation (Equation 1).

### 2.3. Implementing RNAliHiKinetics

The pipeline for kinetic folding simulation consists of three main steps: (1) generating the best *k*
*hishreps* from the folding space, (2) calculating transition rates between the *hishreps* by calculating folding pathways, and (3) analyzing folding kinetics based on the transition rate matrix. Instead of using RNAHeliCes and HiPath2 in the first and second step, in the new version, we use RNAliHeliCes and RNAliHiPath. In the third step, the scripts of RNAliHiKinetics simulate the structural dynamics in time.

### 2.4. Datasets

For the alignment of the Spliced Leader RNAs, we performed a BlastN search with the *Leptomonas collosoma* instance against the NCBI NT database using default parameter settings. From the results, we selected the four sequences with the highest coverage and aligned them with ClustalW2 [13]. The alignment of the Trp-Attenuator sequences, tRNA_10 and the t-box leaders were taken from [8]. All alignments are available in the below mentioned repositories.

### 2.5. Availability

The source code of the programs and the datasets used in this publication are available at: https://github.com/Ibvt/RNAliHeliCes and https://github.com/Ibvt/RNAliHiKinetics.

## 3. Results

### 3.1. Algorithm for Simulating Folding Kinetics of Aligned RNAs

The basic idea of the algorithm is taken from HiKinetics and adopted to work on alignments of related RNAs. For this, we needed to develop an algorithm for the prediction of *hishapes* for a set of aligned sequences and adopt the free energy evaluation of the folding pathway heuristics to work on alignments. For the first, we essentially did the same as described in Voß [8], but with the help of Bellmans GAP [14], which is a framework for the development of dynamic programming algorithms. It offers support for alignments as input, so that we only had to design an algebra for the combined free energy and covariation (pseudo free energy) scoring. The resulting algorithm, called RNAliHeliCes, provides the same helix-center based abstraction levels as described in [15]. As a short reminder, secondary structures consist of five loop types that are closed by helices, namely hairpin, bulge, internal, stacking, and multiple loop. Thus, a helix can be of type hl, bl, il, or ml. Please note that we do not consider stacking loops here because they only elongate helices. We define the position of a helix by its innermost base pair (i,j), more precisely by the central position of the helix, which is $\frac{i+j}{2}$. Additionally, we mark the helix index with *m*, *b*, or *i* for multiple, bulge, or internal loop, respectively. A mapping function π now maps any secondary structure to a list of helix indices that is called helix index shape (*hishape*). In [15], we defined four different levels of abstraction and their corresponding mapping functions: $\pi_{h}$ retains only hairpin loop helices, $\pi_{h+}$ additionally keeps track of the nesting within multiloops, while $\pi_{m}$ and $\pi_{a}$ also keep track of multiloops and all helices, respectively.

RNAliHeliCes supports the same heuristic filtering options that were introduced with RNAHeliCes and we refer the user to the detailed description of these in [15]. Notwithstanding, we want to explicitly mention the (pseudo) free energy filter ‘-x
*e*’, where *e* defines the maximum (pseudo) free energy allowed for a substructure to be inserted in the external loop. This filter resembles the intuition that a substructure in the external loop with non-negative (pseudo) free energy (ΔG≥0) is unlikely to form and can thus be excluded from the search space. Figure 1 shows the results of applying RNAliHeliCes to an alignment of five Spliced Leader (SL) RNAs from different Trypanosomes. It also shows the effect of the pseudo free energy filter (here ‘-x 0’), which would remove the hishapes marked with a ‘*’.

As an example, *hishapes* [38] and [14,38] differ by an additional helix with index 14, which increases the pseudo free energy by 1.54.

The second part that had to be adapted to work on sets of aligned sequences is HiPath2, which computes folding pathways between two *hishapes*. For the new purpose, scores for intermediate structures had to be computed based on the alignment and the combined free energy and covariation scoring (pseduo free energy). Adapting the energy evaulation to alignments was done with the help of functions from the Vienna RNA package 2 [16], which is detailed in the Materials and Methods section.

### 3.2. Comparison of Simulated Kinetics for Aligned and Single Sequences

In order to validate that kinetic analyses for alignments are reasonable, we compared the kinetics computed for a set of aligned sequences with their individual kinetic simulations. To be able to do this, we have to consider that helix indices may be shifted in the aligned sequences compared to the individual ones. This is a result of the alignment procedure that may introduce gaps into the sequences. An example using Spliced Leader (SL) RNA sequences from *Leptomonas collosoma* and (c) *Trypanosoma theileri* is shown in Figure 2.

Here, the *hishape* [27] for the aligned SL RNA sequences corresponds to *hishape* [27] in *L. collosoma* and *hishape* [25] in *T. theileri*.

The SL RNA from *L. collosoma* is considered a conformational switch and the kinetics of the switching have been addressed earlier [17]. For our comparison, we collected four homologs of the *L. collosoma* SL RNA via BLASTn searches, aligned them and grouped the consensus and individual *hishapes* according to Table 1, which also gives the species names of the collected SL RNAs.

We used RNAHiKinetics to simulate the folding kinetics of the individual sequences (Figure 3a–e) and RNAliHiKinetics for the multiple alignment (Figure 3f).

The simulated folding kinetics for the *L. collosoma* SL RNA shown in Figure 3a nicely meet the expectations for a sequence with two similarly stable foldings and are in line with results from more fine-grained simulation methods as we have shown in [5]. In the overall view, it can be seen that all folding kinetics are dominated by either the A (red line) or the B (blue dotted line) group except those of the SL RNA from *Trypanosomatidae* sp. ECU-07 shown in Figure 3b, which lacks a *hishape* of the B group as can be already seen in Table 1. In the remaining four individual folding kinetics (a,c,d,e), the A and the B group occur with a substantial share in at least one time point. Interestingly, there is no single group that dominates in all sequences. While in (c), the A group is more abundant than the B group, in (d) and (e), it is the other way around. This may simply reflect the fact that not the thermodynamically stable structure is functionally important, but rather the flexibility to attain different conformations. The consensus kinetics shown in Figure 3f clearly show the same characteristics as the individual simulations.

### 3.3. Simulated Folding Kinetics for the trp-Attenuator

Attenuation is a well-known regulatory mechanism found in several bacterial operons coding for amino-acid-synthesis genes. It is based on an element of a leader sequence that comprises a termination hairpin and a cluster of codons for the respective amino-acid. If the level of the amino acid is high, ribosomes can quickly pass this cluster and the terminator hairpin persists, resulting in transcriptional termination. On the other hand, if amino acid levels are low, the ribosome pauses at the codon cluster. This allows the mRNA to refold, whereby the terminator hairpin is destroyed and transcription can proceed into the structural genes. This mode of regulation is best studied in *Escherichia coli*, but many other bacteria share this feature. In order to find out if we can detect a signal of this refolding mechanism with the simulation of folding kinetics for aligned RNAs, we took the trp-Attenuator alignment from [8] and computed the consensus folding kinetics.

The first two *hishapes* ([18,93] and [55.5]) computed by RNAliHeliCes agree with two experimentally verified consensus structures (see Figure 4).

Since the bases which form the two stem-loops (red and blue) in the first consensus structure overlap the ones forming the single stem-loop (violet) in the second consensus structure, the two structures are mutually exclusive. Another interesting finding is that the third *hishrep* in comparison to the second has an additional stem loop centered at position 58.5, but scores worse, which is likely the result of less conservation. Due to this, we assume that the *hishapes* [55.5] and [58.5] are functionally similar and we group them for the upcoming interpretation. Figure 5 shows the results of the kinetic simulation with RNAliHiKinetics. The functionally similar *hishapes* [55.5] and [58.5] come up quite early together with the possible transition *hishapes* [18,98], [10,55.5,98], and [10,93], which represent alternative refolding pathways to *hishape* [18,93]. The latter dominates only late in the course of the simulation.

### 3.4. Runtime

In order to analyse the runtime of RNAliHiKinetics, we collected nine alignments of different RNA families with lengths ranging from 56 to 304 nucleotides and simulated folding kinetics based on the top 100 consensus *hishapes*. The results are summarized in Figure 6 and nicely show that the consensus approach of RNAliHiKinetics (RNAliHeliCes + RNAliHiPath) improves efficiency by up to two orders of magnitude compared to RNAHiKinetics (RNAHeliCes + HiPath2). It is important to note that, for the single sequence approach of RNAHiKinetics, most computations had to be restricted to the 100 best *hishapes* (marked with ‘*’ in Figure 6) to achieve acceptable runtime.

The main part of the improvement is in RNAliHeliCes, which is not unexpected because the pathway computation is essentially identical for HiPath2 and RNAliHiPath.

The results also show that the runtime depends on the alignment length, which is expected, but also that it correlates negatively with the mean pairwise identity. This is also not unexpected because the mean pairwise identity impacts the base-pairing possibilities. The less conserved the sequences are, the less base pairs are possible, which results in smaller numbers of secondary structures or, in our case, *hishapes* that have to be evaluated. A similar behaviour is to be expected for the memory consumption.

## 4. Discussion

Here, we present RNAliHeliCes, RNAliHiPath and RNAliHiKinetics, a suite of algorithms for the analysis of conserved secondary structures and their folding kinetics. It is based on a position-aware structure abstraction and uses a combined free energy and covariation scoring based on alignments. The key benefit of this is that the incorporation of information from evolutionary conservation improves the accuracy, while in parallel it dramatically reduces the search space. As a result, the runtime is reduced to up to two orders of magnitude, which makes it possible to study the folding kinetics of long RNAs (>250 nt). On the other hand, large scale simulations of short or medium-sized RNAs become possible.

The validity of the approach is demonstrated by the fact that, for the set of Spliced Leader RNAs, the consensus folding kinetics nicely reflect the bistable character of these RNAs, while it is also similar to the simulated kinetics of the experimentally best studied instance from *L. collosoma*. Nevertheless, the alignment quality and the weighting of the covariance score in computing pseudo free energies may have a significant impact on the simulation results. The latter will especially be the subject to further investigations. Here, we want to emphasize that a benchmarking data set is urgently needed to perform a thorough assessment of the performance.

Furthermore, in the current setting, we use the *hishape* representative structures and their pseudo free energies to simulate folding kinetics. By this, we totally neglect the size of the *hishape* class, which likely has an impact on the kinetics because the larger the class, the more alternative folding pathways are possible. A possible solution to this is to use *hishape* ensemble pseudo free energies computed via a partition function approach. A method that could also be used to improve the simulation accuracy of RNAHiKinetics.

In its current form, RNAliHiKinetics simulates folding kinetics of full-length sequences, which is not always the most realistic scenario. Especially in bacteria, the growing transcript starts to fold before it is fully transcribed, so-called co-transcriptional folding. Here, stable folding intermediates of subsequences can govern the folding process, such that the actually occurring folding trajectories are significantly altered. Modeling this mode of action would require to keep track of the sequence span covered by individual *hishapes* and a time-dependent span extension during the simulation, which would sequentially add accessible *hishapes* to the simulation.

Essentially, RNAliHiKinetics is based on the hypothesis that a superimposed folding space exists among the homologous RNAs. Similar to the prediction of RNA secondary structure using comparative information, the superimposition of folding spaces can be approached in three different ways. First, one can align homologous RNA sequences and then construct the consensus folding space based on this multiple sequence alignment. We call this “Plan A“ in analogy to consensus structure prediction schemes introduced by Gardner and Giegerich [21]. Accordingly, the reverse approach to first construct individual folding spaces and then “align“ them, is called “Plan C“, and “Plan B“ would be to simultaneously align and construct the folding spaces. With respect to this, RNAliHiKinetics implements Plan A. In Plan C, the most difficult step is the alignment of folding spaces, which are high-dimensional and, thus, no common alignment algorithm can be used. At first glance, Plan C might sound the most complicated, but this is essentially a version of the Sankoff algorithm [22] for simultaneous alignment and folding, extended by a non-ambiguous, suboptimal backtracking procedure. Keeping in mind that variants of this algorithm are available for many years and that computational power has dramatically increased since the proposal of the algorithm, it does not seem unrealistic to implement such an algorithm. Combined with abstraction and dedicated filtering methods, it might also be possible to achieve an acceptable runtime and memory footprint.

## Figures and Tables

**Figure 1 genes-12-00347-f001:**
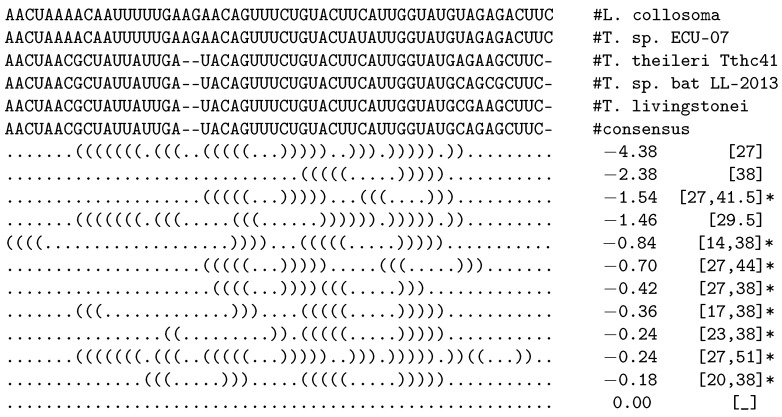
Consensus *hishapes* for the alignment of five SL RNA sequences. *Hishapes* marked with ‘*’ would not show up when using the pseudo free energy filter ‘-x 0’, which does not allow substructures with positive pseudo free energy in the external loop.

**Figure 2 genes-12-00347-f002:**
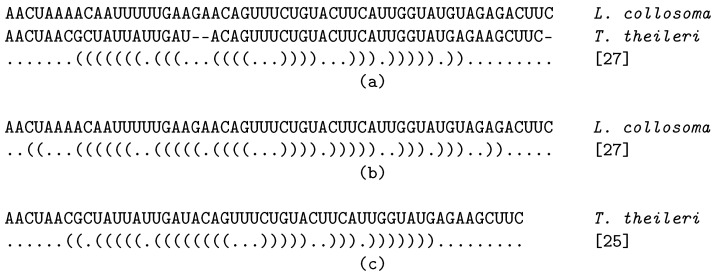
Helix index shift in (**a**) aligned sequences compared to individual sequences of the SL RNAs from (**b**) *L. collosoma* and (**c**) *T. theileri*.

**Figure 3 genes-12-00347-f003:**
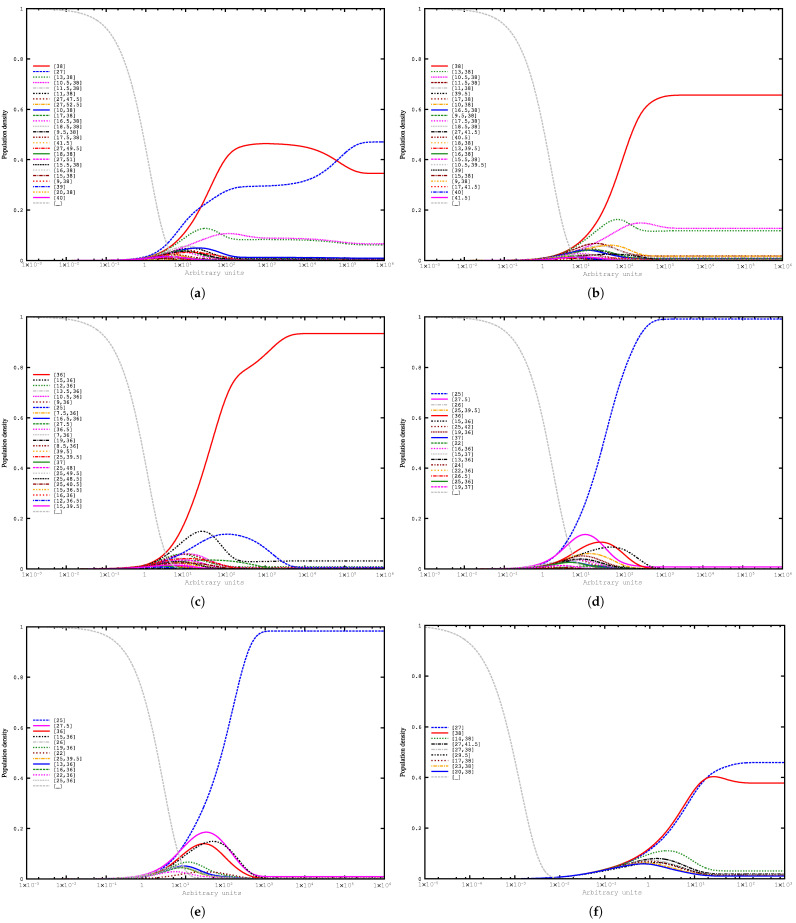
Folding kinetics of (**a**) *Leptomonas collosoma* spliced leader RNA gene repeat unit; (**b**) *Trypanosomatidae* sp. ECU-07 trans-spliced leader sequence SL gene; (**c**) *Trypanosoma theileri* isolate Tthc41 clone 5 trans-spliced leader; (**d**) *Trypanosoma* sp. bat LL-2013 isolate TCC60 clone 1 trans-spliced leader; (**e**) *Trypanosoma livingstonei* isolate TCC1933 clone 2 trans-spliced leader; (**f**) consensus folding space calculated with RNAliHiKinetics based on sequences; (**a**–**e**) above. The sequences (**b**–**e**) were collected by a BlastN search against the nt database with default settings and sequence (**a**) as query.

**Figure 4 genes-12-00347-f004:**
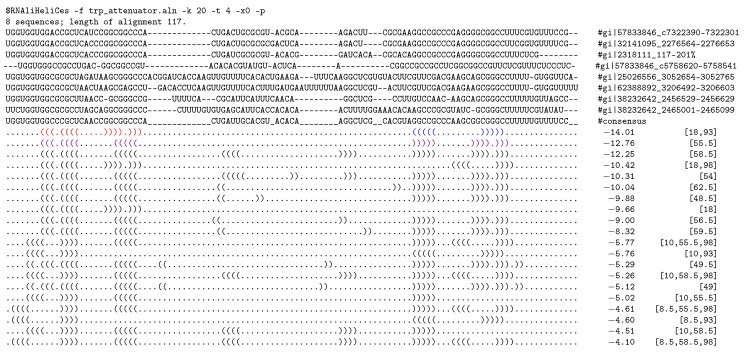
The first 20 *hishapes* of the *trp*-attenuator [18,19,20]. Following the alignment, *hishape* representative structures (*hishreps*) are shown with their pseudo free energy (in kcal/mol) and the corresponding *hishape*.

**Figure 5 genes-12-00347-f005:**
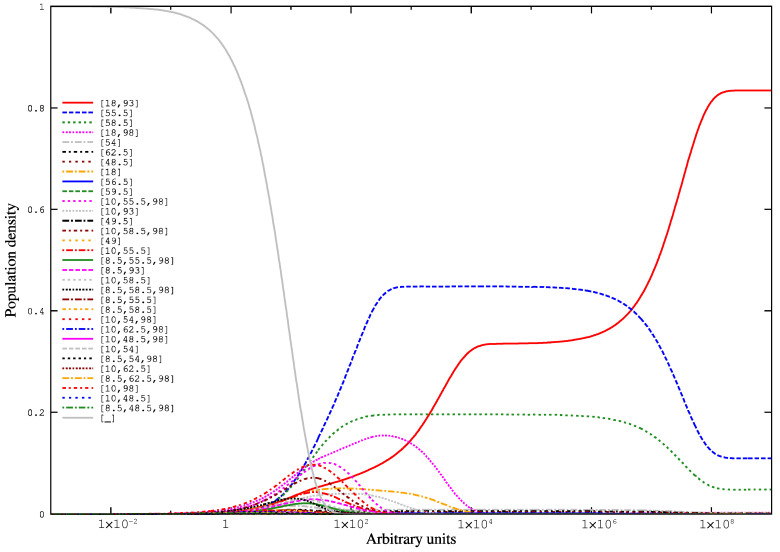
Folding kinetics of the *trp*-attenuator alignment simulated with RNAliHiKinetics. The simulation was based on all strictly negative $\pi_{h}$
*hishapes* plus the open chain ([_]), which was used as the starting structure for this simulation.

**Figure 6 genes-12-00347-f006:**
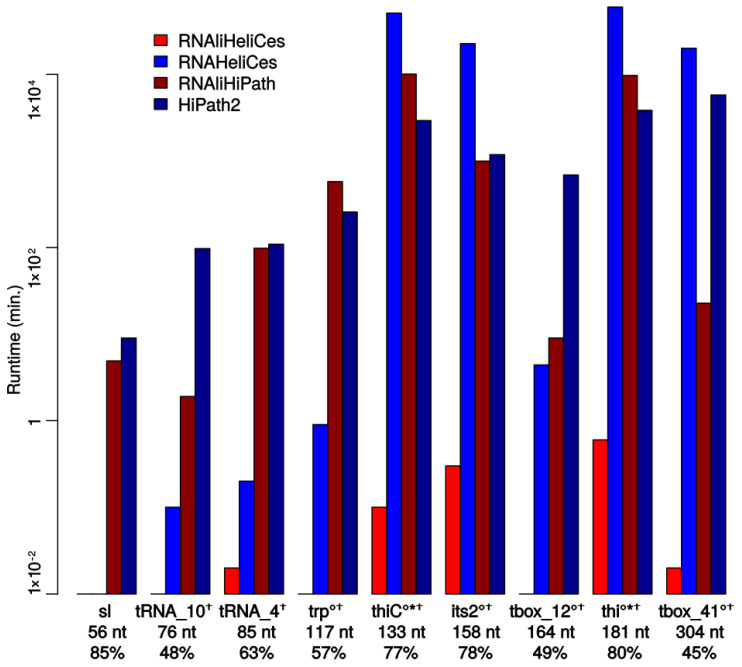
Runtime comparison of RNAliHiKinetics (RNAliHeliCes + RNAliHiPath) and RNAHiKinetics (RNAHeliCes + HiPath2). Run times were measured on a 176x Intel${}^{®}$ Xeon${}^{®}$ Gold 6152 CPU @ 2.10 GHz machine with 768 GB RAM under openSUSE 15.1 (x86_64). For the examples marked with ‘${}^{\circ }$’, only the first 10,000 *hishapes* in RNAHeliCes, for the ones marked with ‘*’, only the pathways between the first 100 *hishapes* in RNAliHiPath, and for the ones marked with ‘†’, only the pathways between the first 100 *hishapes* in HiPath2 were calculated. The numbers below the names refer to the length and the mean pairwise identity of the alignments, respectively.

**Table 1 genes-12-00347-t001:** *Hishape* groups of Spliced Leader RNAs from five different species, namely (1) *Leptomonas collosoma* (2) *Trypanosomatidae* sp. ECU-07 (3) *Trypanosoma theileri* isolate Tthc41 clone 5 (4) *Trypanosoma* sp. bat LL-2013 isolate TCC60 clone 1 and (5) *Trypanosoma livingstonei* isolate TCC1933 clone 2.

Group Name	1	2	3	4	5
A	[38]	[38]	[36]	[36]	[36]
B	[27]		[25]	[25]	[25]
C	[13,38]	[13,38]	[15,36]	[15,36]	[15,36]
D	[10.5,38]	[10.5,38]			
E				[27.5]	[27.5]

## Data Availability

Publicly available datasets were analyzed in this study. This data can be found here: https://github.com/Ibvt/RNAliHiKinetics (accessed on 10 December 2020).

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
