# Peer review of "Simulation of Folding Kinetics for Aligned RNAs"

_genes, 2021, doi:10.3390/genes12030347_

Round 1
Reviewer 1 Report
Huang and Voß presented an article about a new approach named RNAliHiKinetics. The algorithm is responsible for the simulation of folding kinetics, but the main novelty lies in the support for multiple sequence alignment on the input.
The article concerns a significant topic of folding kinetics simulations. Authors did a great job of describing the concept and its importance. They also selected a perfect example later in the manuscript to demonstrate the need for such algorithms. The idea behind RNAliHiKinetics is an adequately designed successor of previous approaches.
Nevertheless, I have some issues to report. I wanted to test the software but failed to do so. First, I tried compiling RNAliHeliCes, but I got errors in line 433 (my g++ version is 10.2.0 if that helps):
rnaalihipath.cc: In function ‘int main(int, char**)’:rnaalihipath.cc:433:20: error: no match for ‘operator==’ (operand types are ‘bool’ and ‘std::fstream’ {aka ‘std::basic_fstream<char>’})
433 | if ((false == ssFileStream) || (false == ssFileStream.is_open()))
| ~~~~~~^~~~~~~~~~~~~~~
I commented out the first component of "if" and the project compiled fine.
BTW. The project contains Valgrind logs committed into the source code files (e.g. https://github.com/Ibvt/RNAliHeliCes/blob/master/src/rnaalihipath.cc), which is a bit odd and uncommon.
Then I tried to follow the tutorial on RNAliHiKinetics GitHub page. The tutorial says that I should run the following command: "RNAliHeliCes RF00500_seed.aln -k 100". However, the RNAliHeliCes project generated two binaries: RNAalihipath and RNAalihishapes. I tried exchanging RNAliHeliCes with RNAalihishapes in the command, but it resulted with "*** stack smashing detected ***: terminated". Maybe the reason was that commented-out "if" clause? I am not sure, and I stopped trying at this point.
Therefore, this article's most significant issue is that it is difficult to compile and run the software. I want the authors to clean up the sources and README files so that there is one unambiguous instruction on building everything from the beginning and running at least a single successful case.
I also have some minor comments regarding the manuscript:
- In Table 1, it is unclear what SUB1, ..., SUB3 mean.
- In Section 2.2, the last paragraph discusses the plots shown in Figure 3. I think it would be good to add a single sentence to the paragraph mentioning that RNAliHiKinetics result is on panel (f). Yes, I know the information is in the caption, but I was a bit confused so that other readers might be too. Maybe the authors could rephrase the first two sentences, e.g.: "We used RNAHiKinetics to simulate the folding kinetics of the individual sequences (Figure 3(a-e)) and RNAliHiKinetics for the multiple alignment (Figure 3(f))".
- In the same paragraph, there is the phrase "all folding kinetics are dominated by the A (red line) and the B (blue line)". Did you mean "... are dominated EITHER by the A (red line) OR the B (blue line)"?
- Later you write: "Interestingly, there is no dominating group for all sequences". I assume you mean "there is no single group that dominates in all sequences"? Can you please rewrite to make it clear?
- In section 2.3, I think you used a mental shortcut writing that the "translation can quickly pass this cluster [of amino acid codons]". It is not the "translation" process that passes the nucleotides, but the ribosome. Or did you have something else in mind? Can you please rephrase this?
- Figure 6 is empty in the reviewer PDF.
- In section 3, did you mean "superposition" instead of "superimposition"?
- Equation (3) is a bit unclear to me. Is the summation going like this: "for k=1 to M: for l=k to M: ..."?
Author Response
Point 1: Huang and Voß presented an article about a new approach named RNAliHiKinetics. The algorithm is responsible for the simulation of folding kinetics, but the main novelty lies in the support for multiple sequence alignment on the input.
The article concerns a significant topic of folding kinetics simulations. Authors did a great job of describing the concept and its importance. They also selected a perfect example later in the manuscript to demonstrate the need for such algorithms. The idea behind RNAliHiKinetics is an adequately designed successor of previous approaches.
Response 1: Thank you very much for this encouraging comment.
Point 2: Nevertheless, I have some issues to report. I wanted to test the software but failed to do so. First, I tried compiling RNAliHeliCes, but I got errors in line 433 (my g++ version is 10.2.0 if that helps):
rnaalihipath.cc: In function ‘int main(int, char**)’:
rnaalihipath.cc:433:20: error: no match for ‘operator==’ (operand types are ‘bool’ and ‘std::fstream’ {aka ‘std::basic_fstream<char>’})
433 | if ((false == ssFileStream) || (false == ssFileStream.is_open()))
| ~~~~~~^~~~~~~~~~~~~~~
I commented out the first component of "if" and the project compiled fine.
Response 2: We have improved the code in line 443 with "if ((ssFileStream.rdstate() & std::ifstream::failbit ) != 0 )".
Point 3: BTW. The project contains Valgrind logs committed into the source code files (e.g. https://github.com/Ibvt/RNAliHeliCes/blob/master/src/rnaalihipath.cc), which is a bit odd and uncommon.
Response 3: We have deleted the Valgrind logs.
Point 4: Then I tried to follow the tutorial on RNAliHiKinetics GitHub page. The tutorial says that I should run the following command: "RNAliHeliCes RF00500_seed.aln -k 100". However, the RNAliHeliCes project generated two binaries: RNAalihipath and RNAalihishapes. I tried exchanging RNAliHeliCes with RNAalihishapes in the command, but it resulted with "*** stack smashing detected ***: terminated". Maybe the reason was that commented-out "if" clause? I am not sure, and I stopped trying at this point.
Response 4: We have no real explanation for the wrong naming of the binaries. The error "*** stack smashing detected ***" can be solved by adding 'CFLAGS="-fno-stack-protector" CPPFLAGS="-std=c++98" CXXFLAGS="-std=c++98 -fno-stack-protector"' to the "./configure" command, which we also added in the README file.
Point 5: Therefore, this article's most significant issue is that it is difficult to compile and run the software. I want the authors to clean up the sources and README files so that there is one unambiguous instruction on building everything from the beginning and running at least a single successful case.
Response 5: We apologize for this inconvinience, but our tests before submission were successful. Nevertheless, we could reproduce the errors on a different computer and improved the documentation. Furthermore, we want to emphasize that the project is hosted on github, such that users can easily report issues.
Point 6: I also have some minor comments regarding the manuscript:
In Table 1, it is unclear what SUB1, ..., SUB3 mean.
Response 6: We wanted to express that they are of minor (sub) importance, but we agree that this is confusing. We changed the names to C, D and E, respectively.
Point 7: In Section 2.2, the last paragraph discusses the plots shown in Figure 3. I think it would be good to add a single sentence to the paragraph mentioning that RNAliHiKinetics result is on panel (f). Yes, I know the information is in the caption, but I was a bit confused so that other readers might be too. Maybe the authors could rephrase the first two sentences, e.g.: "We used RNAHiKinetics to simulate the folding kinetics of the individual sequences (Figure 3(a-e)) and RNAliHiKinetics for the multiple alignment (Figure 3(f))".
In the same paragraph, there is the phrase "all folding kinetics are dominated by the A (red line) and the B (blue line)". Did you mean "... are dominated EITHER by the A (red line) OR the B (blue line)"?
Later you write: "Interestingly, there is no dominating group for all sequences". I assume you mean "there is no single group that dominates in all sequences"? Can you please rewrite to make it clear?
Response 7: Yes, the reviewer is right. We changed the paragraphs according to the reviewers suggestion.
Point 8: In section 2.3, I think you used a mental shortcut writing that the "translation can quickly pass this cluster [of amino acid codons]". It is not the "translation" process that passes the nucleotides, but the ribosome. Or did you have something else in mind? Can you please rephrase this?
Response 8: Thanks for pointing this out. The reviewer is right. This formulation was sloppy and we rephrased it.
Point 9: Figure 6 is empty in the reviewer PDF.
Response 9: We sincerely apologise for this error. Figure 6 was included in the latex sources but, for reasons we cannot explain, was not inserted into the document when it was compiled by the submission system. Nevertheless, we should have checked the PDF proof more carefully.
Point 10: In section 3, did you mean "superposition" instead of "superimposition"?
Response 10: No and yes. Both terms have the same meaning.
Point 11: Equation (3) is a bit unclear to me. Is the summation going like this: "for k=1 to M: for l=k to M: ..."?
Response 11: Yes, but there was an error in the equation such that "l=k" was included. We have corrected the equation to exclude this.
Reviewer 2 Report
The manuscript "Simulation of Folding Kinetics for aligned RNAs" by J. Huang and B. VoB describes a set of programs that uses RNA sequence alignments to determine the folding kinetics given a set of evolutionarily related RNA sequences. In particular, their algorithms seem to be capable of providing indications of dominant structures in RNAs that may have more than one folding state. The algorithms utilize concepts of structure abstraction and evolutionary conservation amongst the aligned sequences to locate common parts of the folding spaces related to the set of aligned sequences. Part of the rationale is to reduce the combinatorics of such analyses by eliminating structures from consideration that do not pass a defined threshold for energy improvement. The results indicate good agreement for a couple of examples. The paper is well written, but could be improved if the following suggestions are included:
1) Even though the examples provided seem to correlate with previous results, it would be helpful if the authors tested their programs on more examples, e.g. sets of riboswitches to more thoroughly show the accuracy of their predictions.
2) The authors should provide somewhat more discussion on issues related to co-transcriptional folding and the use of the alignments. There is a reference to KINFOLD in this regard and the authors allude to their own ability to obtain folding trajectories utilizing the sequence alignments. But, more information would be helpful on how their algorithms capture results inherent in co-transcriptional folding vs. full folding sequence folding. This seems to be especially important given that their method seems to rely on an energy filtering algorithm, which utilizes full sequence affects rather than local affects.
3) Figure 6 seems to be missing even though a caption is provided.
Author Response
The manuscript "Simulation of Folding Kinetics for aligned RNAs" by J. Huang and B. VoB describes a set of programs that uses RNA sequence alignments to determine the folding kinetics given a set of evolutionarily related RNA sequences. In particular, their algorithms seem to be capable of providing indications of dominant structures in RNAs that may have more than one folding state. The algorithms utilize concepts of structure abstraction and evolutionary conservation amongst the aligned sequences to locate common parts of the folding spaces related to the set of aligned sequences. Part of the rationale is to reduce the combinatorics of such analyses by eliminating structures from consideration that do not pass a defined threshold for energy improvement. The results indicate good agreement for a couple of examples. The paper is well written, but could be improved if the following suggestions are included:
Point 1: Even though the examples provided seem to correlate with previous results, it would be helpful if the authors tested their programs on more examples, e.g. sets of riboswitches to more thoroughly show the accuracy of their predictions.
Response 1: We can fully understand the reviewers request. Unfortunately, there is no objective benchmark data set available to do this. Therefore, we decided to add a sentence to the discussion to mention this issue.
Point 2: The authors should provide somewhat more discussion on issues related to co-transcriptional folding and the use of the alignments. There is a reference to KINFOLD in this regard and the authors allude to their own ability to obtain folding trajectories utilizing the sequence alignments. But, more information would be helpful on how their algorithms capture results inherent in co-transcriptional folding vs. full folding sequence folding. This seems to be especially important given that their method seems to rely on an energy filtering algorithm, which utilizes full sequence affects rather than local affects.
Response 2: Yes, good point. We added a paragraph about co-transcriptional folding in the discussion.
Point 3: Figure 6 seems to be missing even though a caption is provided.
Response 3: We sincerely apologise for this error. Figure 6 was included in the latex sources but, for reasons we cannot explain, was not inserted into the document when it was compiled by the submission system. Nevertheless, we should have checked the PDF proof more carefully.
Reviewer 3 Report
The paper presents a new pipeline of algorithms for the analysis of conserved RNA secondary structures and their folding kinetics. In the pipeline the second algorithm, namely RNALIHIPATH, is new and is the main contribution of the paper.
The new pipeline has a benefit on the accuracy of the analysis performed with respect to existing approaches. This is shown by suitable examples and clearly discussed. I have no comments about this part.
The authors also claim that there is an important improvement (two orders of magnitude) on the efficiency of the computational time, so longer sequences may be analysed in a reasonable time. Unfortunately, Figure 6, which is the one showing this improvement, is missing in the pdf and I could not easily recover it in the github repositories that are linked. So I cannot judge about this improvement. The picture must be given in a new version and also the "two order of magnitude" improvement should be detailed better: with respect to what exactly?
I expect the authors resubmit a new version in which they fix the figure issue and explain better the runtime improvement part.
Author Response
The paper presents a new pipeline of algorithms for the analysis of conserved RNA secondary structures and their folding kinetics. In the pipeline the second algorithm, namely RNALIHIPATH, is new and is the main contribution of the paper.
The new pipeline has a benefit on the accuracy of the analysis performed with respect to existing approaches. This is shown by suitable examples and clearly discussed. I have no comments about this part.
Point 1: The authors also claim that there is an important improvement (two orders of magnitude) on the efficiency of the computational time, so longer sequences may be analysed in a reasonable time. Unfortunately, Figure 6, which is the one showing this improvement, is missing in the pdf and I could not easily recover it in the github repositories that are linked. So I cannot judge about this improvement. The picture must be given in a new version and also the "two order of magnitude" improvement should be detailed better: with respect to what exactly?
I expect the authors resubmit a new version in which they fix the figure issue and explain better the runtime improvement part.
Response 1: We sincerely apologise for this error. Figure 6 was included in the latex sources but, for reasons we cannot explain, was not inserted into the document when it was compiled by the submission system. Nevertheless, we should have checked the PDF proof more carefully. It is now included and we also improved the runtime part.
Round 2
Reviewer 2 Report
The authors have satisfactorily addressed the issues raised in the last review.